# The Susceptibility of Chickens to Zika Virus: A Comprehensive Study on Age-Dependent Infection Dynamics and Host Responses

**DOI:** 10.3390/v16040569

**Published:** 2024-04-07

**Authors:** Ruth H. Nissly, Levina Lim, Margo R. Keller, Ian M. Bird, Gitanjali Bhushan, Sougat Misra, Shubhada K. Chothe, Miranda C. Sill, Nagaram Vinod Kumar, A. V. N. Sivakumar, B. Rambabu Naik, Bhushan M. Jayarao, Suresh V. Kuchipudi

**Affiliations:** 1Department of Veterinary and Biomedical Sciences, Pennsylvania State University, University Park, PA 16802, USA; ruthn@psu.edu (R.H.N.); levinalimhs@gmail.com (L.L.); mrk5415@psu.edu (M.R.K.); ian.bird@jhuapl.edu (I.M.B.); glb5150@psu.edu (G.B.); bmj3@psu.edu (B.M.J.); 2DermBiont, Inc., 451 D Street, Suite 908, Boston, MA 02210, USA; 3Applied Biological Sciences Group, The Johns Hopkins University Applied Physics Laboratory, Laurel, MD 20723, USA; 4College of Medicine, Pennsylvania State University, Hershey, PA 17033, USA; 5Department of Infectious Diseases and Microbiology, University of Pittsburgh, Pittsburgh, PA 15261, USA; sougat.misra@pitt.edu (S.M.); shc319@pitt.edu (S.K.C.); 6Department of Biology, Pennsylvania State University, University Park, PA 16802, USA; miranda.sill@jefferson.edu; 7College of Veterinary Science, Sri Venkateswara Veterinary University, Tirupati 517 602, Andhra Pradesh, India; nagaram_vinod@yahoo.com (N.V.K.); priyamshiva@gmail.com (A.V.N.S.); dr.babyrambabu@gmail.com (B.R.N.); 8Center for Vaccine Research, University of Pittsburgh, Pittsburgh, PA 15261, USA

**Keywords:** Zika virus, chicken, innate immune response, STAT2, antibody, host restriction

## Abstract

Zika virus (ZIKV) remains a public health concern, with epidemics in endemic regions and sporadic outbreaks in new areas posing significant threats. Several mosquito-borne flaviviruses that can cause human illness, including West Nile, Usutu, and St. Louis encephalitis, have associations with birds. However, the susceptibility of chickens to ZIKV and their role in viral epidemiology is not currently known. We investigated the susceptibility of chickens to experimental ZIKV infection using chickens ranging from 1-day-old chicks to 6-week-old birds. ZIKV caused no clinical signs in chickens of all age groups tested. Viral RNA was detected in the blood and tissues during the first 5 days post-inoculation in 1-day and 4-day-old chicks inoculated with a high viral dose, but ZIKV was undetectable in 6-week-old birds at all timepoints. Minimal antibody responses were observed in 6-week-old birds, and while present in younger chicks, they waned by 28 days post-infection. Innate immune responses varied significantly between age groups. Robust type I interferon and inflammasome responses were measured in older chickens, while limited innate immune activation was observed in younger chicks. Signal transducer and activator of transcription 2 (STAT2) is a major driver of host restriction to ZIKV, and chicken STAT2 is distinct from human STAT2, potentially contributing to the observed resistance to ZIKV infection. The rapid clearance of the virus in older chickens coincided with an effective innate immune response, highlighting age-dependent susceptibility. Our study indicates that chickens are not susceptible to productive ZIKV infection and are unlikely to play a role in the ZIKV epidemiology.

## 1. Introduction

*Orthoflavivirus zikaense*, commonly known as Zika virus (ZIKV), is an arthropod-borne small enveloped positive-stranded RNA virus belonging to the Flaviviridae family [1]. Initially identified in 1947, intensified scrutiny of the virus followed a surge in microcephaly cases associated with a 2015 ZIKV outbreak in Brazil [2]. The significant outbreak and the severity of linked long-term health consequences, including congenital Zika syndrome (CZS), led the World Health Organization (WHO) to declare a Public Health Emergency of International Concern on 1 February 2016 [3,4,5,6]. While reported cases of ZIKV virus disease globally declined starting in 2017, it is crucial to note that Zika virus transmission persistently continues at lower levels in various countries across the Americas and other endemic regions [7]. Alarmingly, in 2019, the first instances of Zika virus disease locally transmitted by mosquitoes were reported in Europe, signifying an ongoing threat. Furthermore, in 2021, ZIKV outbreaks were identified in India [8], emphasizing the persistent and global nature of this health concern [9,10,11].

ZIKV is primarily transmitted through the bite of a mosquito. Several flaviviruses that are transmitted by various species of *Culex* mosquitoes and can cause human illness are associated with birds, including West Nile virus (WNV), Usutu virus, and St. Louis encephalitis virus [12,13,14]. To date, no *Culex* species has been unequivocally confirmed as a competent vector for ZIKV transmission [15,16,17,18,19,20,21]. The two major vectors of ZIKV for humans, *Aedes aegypti* and *Aedes albopictus*, have a wide and expanding geographical range across all continents except Antarctica [22,23,24]. Although they have feeding preferences for mammals, these mosquitos are known to forage on avian species, including chickens [25,26,27,28,29]. Chicken-originated bloodmeals have been identified sporadically in *Ae. aegypti* captured in urban settings in Africa and Asia, appearing in 0.7 to 1.4% of tested mosquitoes [29,30]. *Ae. albopictus* captured in urban and suburban locations in North and South American had similar rates (1.5 to 1.7%) [26,31]. In one study, *Ae. aegypti* held bloodmeals with mixed human and chicken origins [29]. Domestic chickens are kept by humans throughout the world, ranging from small backyard flocks to large farms. Close contact with humans and exposure to potential and known vectors of ZIKV have raised the question about susceptibility of chickens to ZIKV. If chickens are susceptible to ZIKV, the possibility for ZIKV transmission from chickens to humans could also exist, albeit with very low risk.

A high proportion of ZIKV infections are asymptomatic in humans, as suggested by a high seroprevalence in communities despite low case numbers [32]. Since 2007, ZIKV outbreaks have been more commonly reported, aided in part by improved specificity of serological assays and molecular diagnostics [33,34,35]. Most symptomatic patients exhibit mild symptoms, such as general malaise, low fever, and myalgia. Starting in 2013, ZIKV infections became increasingly associated with neurological symptoms and fetal development disruptions [2,36,37,38,39]. Patients reported experiencing Guillan Barre’ syndrome, which is characterized by progressive weakness caused by damage to peripheral nerves [39,40]. CZS is the most concerning condition associated with Zika virus infection. When mothers are infected during pregnancy, vertical transmission of ZIKV to the fetus can cause direct neurological damage and loss of intracranial volume [38,41,42]. Clinical features of CZS can include microcephaly, ocular and aural abnormalities, and musculoskeletal dysfunction. CZS is associated with delayed neurodevelopment and reduced life expectancy in affected children [43,44,45,46,47,48].

ZIKV is maintained via a sylvatic cycle and re-emerges periodically in endemic regions when the mosquito-transmitted virus is acquired by a human [49]. Human–mosquito–human transmission subsequently drives local sustained transmission, although very rare direct human-to-human sexual transmission is also possible [50]. Several non-human primates likely serve as reservoirs or amplifying host species [49,51,52]. Non-human primates shown to be susceptible to experimental infection include native species of Africa, Asia, and the Americas [53,54,55,56,57,58,59]. Other immunocompetent animals shown to support ZIKV infection include some guinea pigs and Syrian golden hamsters [60,61]. Fetuses and newborn pups of immunocompetent mice are also susceptible [62,63]. In susceptible animal models, acute irritability, lethargy, and fever are common, but often infection is not accompanied by clinical signs [56,57,58]. Viremia is typically observed for 2 to 10 days on average after inoculation [53,54,55,56,57,58,59]. In some animal models, viral RNA persists in saliva, urine, feces, and other body fluids for weeks or months [53,54,55,56,57,58,59]. ZIKV is detectable by RT-PCR in the spleen, the brain, and several other tissues [53,54,55,56,57,58,59,60,61,62,63]. In contrast, non-susceptible species have little or no virus detectable in their blood during the first three days after inoculation, and ZIKV is not detectable in their tissues [64,65,66].

Exploration of ZIKV infection in birds is limited. A few historical studies reported antibodies indicating previous natural infection of wild birds [67,68], and ZIKV-specific antibodies were detected in a domestic chicken in 2017 [69]. Embryonated chicken eggs have been known to support ZIKV growth since 1952 [70,71,72,73], and a chicken embryo fibroblast cell line is permissive to ZIKV [74,75]. Studies in developing chicken embryos have found that experimental ZIKV infection results in productive infection and developmental abnormalities. Two studies have examined small groups of juvenile and adult chickens for viremia and seroconversion following ZIKV inoculation [65,66]. Limited tissue sampling was conducted when the animals were euthanized 3 weeks after inoculation. Neonatal mice succumb to ZIKV infection, while older mice are resistant to infection [62,76], and similarly, 1-day-old chicks are more susceptible to WNV than 1-week-old chickens [77]. A large-scale study of chicken susceptibility to ZIKV is still lacking, and the susceptibility of neonatal chickens is unexplored. Through a series of experimental cohorts, we tested eight cohorts of chickens of various ages to measure their susceptibility, identify clinical features and pathology, better understand seroconversion, and observe factors that may affect their vulnerability to ZIKV infection.

## 2. Materials and Methods

### 2.1. Animals

All animal studies were approved by the Pennsylvania State University Institutional Animal Care and Use Committee, protocol 47484. Day-old straight-run chicks and 5–6-week-old pullets (white Leghorn) were purchased from a local approved egg-laying poultry operation. For some experiments on day-old animals, chicks were hatched in the University vivarium from fertilized eggs purchased from the same operation. Chickens were housed in heated negative-pressure HEPA-filtered poultry isolators (PlasLabs) in an ABSL-2 facility and were provided fresh water and commercial poultry feed (DuMor 20% chick starter/grower) ad libitum.

### 2.2. Virus Propagation, Inactivation, and Quantification

ZIKV PRVABC59 (Human/2015/Puerto Rico) was obtained from BEI Resources and was propagated in Vero cells (ATCC CCL-81). Virus-containing supernatants were collected from inoculated cells after 5–7 days of culture, centrifuged for 10 min at 3200× *g*, and aliquoted and stored at −80 °C until use. Infectivity was assessed by 50% tissue culture infectious dose on Vero cells and quantified using the method of Reed and Muench [78]. Killed ZIKV was generated by exposing the virus to UV radiation for 1 h before aliquoting and freezing. Inactivation was confirmed by three blind passages in Vero cells.

### 2.3. Infection Experiments

All animal infection studies were conducted in BSL-2 facilities and approved by the Penn State University Institutional Animal Care and Use Committee (IACUC), protocol 47484. Chickens were inoculated with an injection of 50 μL in the lateral neck with a 25G needle using either subcutaneous or intravenous (jugular vein) inoculation. Virus was diluted to the appropriate concentration in DMEM before inoculation, and mock-infected birds received DMEM. Birds were monitored for the following clinical signs: lethargy, decreased appetite, conjunctivitis, postural abnormalities suggestive of joint and/or muscle pain, weight loss, and neurological signs. In some experiments, cloacal and tracheal swabs were collected and placed in brain heart infusion broth before storing at −80 °C. In some experiments, blood samples were collected from the jugular vein. At designated timepoints, birds were euthanized, and blood and tissues were collected. Blood samples were collected into uncoated (for serum) or K2 EDTA-coated (for plasma) Vacuette tubes (Greiner Bio-One, Monroe, NC, USA). Serum (after clotting) or plasma were separated from blood samples by centrifugation at 1500× *g* at 4 °C for 20 min, then stored at −80 °C. Tissues were placed in RNALater, stored for 24 h at 4 °C, removed from RNALater, and stored at −80 °C.

### 2.4. Viral RNA Extraction

Viral RNA was extracted from tissue homogenates, swabs, and plasma using Applied Biosystems MagMAX Pathogen RNA/DNA kit. Tissues were combined with phosphate-buffered saline (20% *w*/*v*) and homogenized in M tubes using the RNA 2.1 program on a gentleMACS Dissociator. Tissue homogenates were then centrifuged at 1000× *g* for 5 min at 4 °C, and the supernatants were used for RNA extraction following the kit’s whole-blood protocol. The low-cell content protocol was followed for plasma, serum, vitreous humor, and tracheal and cloacal swabs. Extracted viral RNA was stored at −80 °C.

### 2.5. Total RNA Extraction from Spleen

To extract total RNA, tissue homogenates from spleens (100 μL) were combined with 600 μL Buffer RLT Plus (Qiagen, Germantown, MD, USA) and passed through a QIAshredder spin column according to the manufacturer’s instructions. The flow through was processed with the RNeasy Plus mini kit (Qiagen). Recovered total RNA was quantified with a NanoDrop Lite (Thermo Fisher Scientific, Waltham, MA, USA) and stored at −80 °C until use.

### 2.6. Host Gene qPCR

The qScript cDNA Synthesis Kit (Quantabio, Gaithersburg, MD, USA) was used to produce cDNA from 500 ng of total RNA per sample according to the manufacturer’s instructions. Quantitative PCR for host gene amplification was performed with the Power SYBR Green Master Mix (Thermo Fisher Scientific, Waltham, MA, USA) using a 1:10 dilution of cDNA and previously described primer sets [79,80,81,82], as shown in Appendix A. The PCR reaction was run at 95 °C for 10 min followed by 40 cycles of 95 °C for 15 s and 60 °C for 60 s. Melt curve analysis was performed for each PCR run to ensure a single peak indicating a single amplicon resulting from the reaction. Using BestKeeper [83], 18S was selected as the house-keeping gene for relative quantification of gene expression using the method of Pflaffl [84]. Differential expression of genes was determined by calculating the relative expression value ratio of each ZIKV-infected animal to the mean of mock-infected animals at the corresponding timepoint. Welch’s two-tailed *t*-test was conducted on data from the two age groups at each timepoint for each gene to determine *p*-values.

### 2.7. Viral qRT-PCR

Detection of ZIKV RNA was performed using SuperScript III Platinum One-Step qRT-PCR Kit on an Applied Biosystems 7500 Fast Real-Time PCR machine on extracted viral RNA from plasma, tissue samples, and swabs using the primers and probe listed in Appendix A [85]. The RT-PCR reaction was run at 50 °C for 30 min, 95 °C for 2 min, and 45 cycles of 95 °C for 15 s and 55 °C for 30 s. For relative quantification of ZIKV viral RNA, 100 μL of virus at 4.22 × 10^7^ TCID_50_/mL was extracted following the procedure for tissue homogenates. This standard RNA was serially diluted 10-fold in nuclease-free water and used in the qRT-PCR reactions with unknown samples.

### 2.8. Serological Assays

An in-house enzyme-linked immunosorbent assay (ELISA) was used to detect anti-ZIKV chicken antibodies in serum. Approximately 5275 TCID_50_ units of ZIKV PRVABC59 diluted in bicarbonate/carbonate buffer (pH 9.4) were coated on a Nunc MaxiSorp 96-well plate and incubated overnight at 4 °C. After washing with wash buffer (Dulbecco’s phosphate-buffered saline (DPBS) with 0.05% TWEEN 20) and overnight incubation with blocking buffer (wash solution with 2% bovine serum albumin (BSA)), serum samples diluted 1:50 in blocking buffer were tested in triplicate and incubated for 1 h at room temperature. Plates were washed, incubated with alkaline phosphatase (AP)-labeled rabbit anti-chicken IgY (Invitrogen, 1:20,000 dilution) for 1 h at room temperature, washed, and developed using the Pierce pNPP Substrate Kit for 30 min before reading absorbance at 405 nm on a Synergy HTX plate reader (BioTek). Using a subset of samples, the performance of the in-house ELISA was compared with the previously-described ZIKV-specific ELISA and plaque reduction neutralization assay (PRNT) [86]. Although results were consistent between the two ELISA assays, no neutralizing antibodies were detected by PRNT.

## 3. Results

### 3.1. ZIKV Does Not Create a Productive Infection in Juvenile Chickens

To investigate if chickens could be susceptible to ZIKV, we performed experimental inoculation of commercial layer chicks (4 days old) and pullets (6 weeks old). Animals were inoculated with 10^5^ TCID_50_ of ZIKV PRVABC59 (Asian lineage) subcutaneously (SQ) in the lateral neck. At 5 and 10 days following inoculation, no plasma viremia was detected in the chickens (Figure 1A,D,G). Throughout the study period, no virus was detected in samples from oropharyngeal swabs and cloacal swabs that were collected daily. Chickens did not exhibit any clinical signs associated with ZIKV infection. No ZIKV was detected by RT-PCR in the brain, spleen, liver, kidney, pancreas, lung, eye, or duodenum.

In a second experiment, 4-day-old chicks and 1-day-old chicks were inoculated intravenously in the external jugular vein with 10^7^ TCID_50_ ZIKV PRVABC59, and viral RNA in plasma was measured from animals sacrificed daily from days 1–5 post-inoculation (Figure 1B). This high dose of ZIKV corresponds to the maximum level which has been shown to be transmitted in experimental mosquito feeding studies [15]. On each of the 5 days after inoculation into 1-day-old chicks, ZIKV RNA was detectable in the plasma of 50% or more of tested chicks (Figure 1E). At 3 days post-inoculation (dpi), ZIKV RNA was detected in the plasma of only 50% of 4-day-old chicks and in 25% of chicks at 5 dpi (Figure 1H). This result in 4-day-old chicks with 10^7^ TCID_50_ was very similar to that from the previous cohort of 4-day-old chicks inoculated with 10^5^ TCID_50_, in which no ZIKV was detected in the plasma at 5 dpi. Overall, these results suggest a dose-dependent and age-dependent susceptibility to ZIKV in chickens, with differences observed between 1-day- and 4-day-old chicks. However, in both age groups, no clinical signs were observed. The finding for 1-day-old chicks was confirmed in a third experiment with a larger number of individuals (Figure 1C); viral RNA was detectable in the plasma of a majority of these chicks at both 3 and 5 dpi (Figure 1F).

Since clinical signs of ZIKV infection in humans can take up to two weeks to manifest, additional cohorts of chickens were observed for a longer period following SQ inoculation with 10^5^, 10^6^, or 10^7^ TCID_50_ of ZIKV PRVABC59 (Figure 2). Animals of 1 day old, 4 days old, and 6 weeks old were followed for up to 16–23 days post-inoculation. Throughout this longer period, no clinical signs or gross pathologies were observed in any age group. In the 6-week-old chickens, no virus was detected in the plasma (Figure 2D) or tissues at early (day 1–3 post-inoculation) or late (days 7 and 16 post-inoculation) timepoints. Chicks inoculated at 4 days old with up to 10^7^ TCID_50_ ZIKV had no detectable virus in the plasma (Figure 2E) or tissues at 14 or 21 days post-inoculation. Similar results were found in chicks inoculated at 1 day old at days 16 and 23 post-inoculation, despite verification that inoculation with 10^7^ TCID_50_ resulted in plasma viremia at 2 dpi (Figure 2F). At 2 dpi, ZIKV RNA was detected in crop, liver, brain, kidney, and spleen of 1-day-old chicks, but not in the duodenum, heart, lung, or eye tissue of the same individuals (Table 1).

Long-term shedding of ZIKV in body fluids, including urine, stool, tears, and saliva, is common in ZIKV infections in humans and monkeys. Tracheal and cloacal swabs collected at 23–34 dpi from chickens inoculated at 1 day old tested negative for ZIKV RNA. Equivalent samples collected at 16 dpi in chickens inoculated at 6 weeks old also tested negative for ZIKV RNA, as did eye tissue and vitreous humor. These results suggest that ZIKV is not perpetuated in the reservoir tissues of chickens.

ZIKV-specific serum IgY levels from ZIKV-infected animals were indistinguishable from those of mock-infected chickens, except in 4-day-old and 1-day-old chicks at 21–23 dpi, where a slight increase in antibodies was observed (Figure 2G–I).

Taken together, these results indicate that juvenile chickens quickly clear ZIKV so that no productive infection is produced. The results also indicate that chicks are potentially susceptible to transient productive infections which are rapidly cleared.

### 3.2. ZIKV Characterization in 1-Day-Old Chickens

A peak in ZIKV RNA was observed at day 1 after inoculation, followed by a subsequent decline, so it was unclear from grouped data whether this observation represented true infection with ZIKV or simply a decay of input virus following the large inoculum dose. To explore this further, a cohort of six chicks was inoculated IV with ZIKV at one day old, and plasma was collected every two days (Figure 3A). Several chicks had peak viremia at 2 days post-inoculation (Figure 3B). The volume of plasma obtained and tested from chick Ha-22 on 2 dpi was 67–88% less than from other chicks, which may explain why this chick had undetectable virus at 2 dpi but detectable virus at 4 dpi (Figure 3B). On one or more days, the serum ZIKV-specific IgY from all ZIKV-inoculated chicks in this tracked cohort was at least 20% higher than that in all mock-inoculated chicks at the same timepoint. The average fold-change in anti-ZIKV serum IgY from ZIKV-infected chicks was significantly higher than from mock-infected controls through 21 dpi (Figure 3).

Further indication of a transient low-level productive infection of ZIKV in young chicks came from analyzing the ZIKV RNA levels in tissues from another cohort of chicks inoculated with 10^7^ TCID_50_ ZIKV PRVABC59 at 1 day old. ZIKV RNA was detected in the brain, spleen, crop, kidney, duodenum, and liver at 1 dpi (Figure 4, Appendix A). Productive viral replication in tissues was indicated by an increase in viral RNA in the duodenum, liver, brain, spleen, and crop between days 2 and 3. Furthermore, animals inoculated with an equivalent amount of inactivated ZIKV showed no detectable ZIKV RNA in any tissues or plasma at any timepoint. No viral RNA was detected at any timepoint in the pancreas tissue. ZIKV RNA was detected in the plasma up to 5 dpi but was undetectable (<58 TCID_50_/mL) by 7 dpi. ZIKV RNA could be detected in the spleen, brain, and crop of at least one chick up to 10 dpi but was undetectable (<53–319 TCID_50_/spleen, <1755–2792 TCID_50_/brain, and <approx. 771–1409 TCID_50_/crop) by 14–16 dpi.

### 3.3. Differential Gene Expression

To characterize the innate immune response to ZIKV in chickens, we examined the expression of a selected set of innate immune response genes and compared the gene expression levels of ZIKV-inoculated chickens with those of uninfected controls. In the spleens of chickens infected at 6 weeks of age, the initial response at 1 dpi was characterized by reduced expression of antiviral genes OAS-L and MDA5 and type-1 interferons IFN-β and IFN-α (Figure 5), which could represent migration of immune cells from the spleen to peripheral sites. By day 3, these genes, as well as proinflammatory IL1β, were highly upregulated compared with mock-infected animals. Interferons, IL1β, and MDA5 remained highly upregulated through 14 dpi. In contrast, spleens from 1-day-old chicks inoculated with ZIKV showed equivalent or slightly downregulated expression of these genes throughout the study period. A much greater level of OAS-L expression was observed at 1 dpi, and OAS-L expression was strongly downregulated at 3 dpi compared with mock-infected controls.

Together, these results indicate an activated innate antiviral gene response in older chickens, which led to rapid control of ZIKV. In contrast, young chickens did not mount a strong antiviral response, thus allowing ZIKV to persist and replicate for multiple days before adaptive immune responses enabled virus clearance.

## 4. Discussion

This study adds to body of evidence indicating that chickens are not susceptible to ZIKV infection. The present study demonstrates that chicks between the ZIKV-susceptible (unhatched) and resistant (post-natal) periods can experience a viremic but unproductive infection when administered a high dose of ZIKV. Previous studies on chickens inoculated with the ZIKV-related flavivirus WNV also demonstrated age-dependent susceptibility to infection. While WNV typically does not cause clinical disease in poultry, they do serve as a sentinel species for the virus [87]. Susceptibility to WNV was also found to vary between different ages of young chickens, with distinct outcomes of inoculation with WNV observed between 1-day- and 1-week-old chicks. In one study, older chicks demonstrated lower viral loads and no clinical signs, whereas younger chicks displayed severe neurological signs and died within as little as 5 dpi [77]. Similar results have been observed in mice with ZIKV: following the neonatal period (defined as 7 days), the lethality and clinical signs associated with ZIKV infection are dramatically reduced, as shown by Li et al. [62].

In susceptible hosts, ZIKV NS5 protein causes NLRP3 inflammasome activation and subsequent production of IL-1β, which can promote an antiviral immune state [88,89,90]. A major driver of host restriction to ZIKV susceptibility is STAT2, which is targeted for degradation by ZIKV NS5 [61,91,92,93]. Loss of STAT2 activity causes inhibition of type I interferon and interferon-stimulated genes. The chicken STAT2 protein sequence is divergent from human and non-human primate STAT2 (Appendix A). Only one-third of the amino acid residues involved in human STAT2 binding to ZIKV NS5 are shared by chicken STAT2 [94] (Appendix A). The majority of the mismatched amino acids (14 residues) are located in the coiled-coil domain (CCD), which is indispensable for the NS5-directed targeting of human STAT2 for proteasomal degradation [95]. Replacement of human STAT2 CCD with mouse CCD prevented targeting of the chimeric STAT2 for degradation during ZIKV infection [95]. Chicken STAT2 CCD has low homology with human CCD, which is reflective of the overall low homology between the two species’ STAT2 amino acids and overall structure, as predicted by AlphaFold [96] (Appendix A). It is therefore likely that chicken STAT2 is not targeted by NS5. This may explain the type I interferon response observed in 6-week-old chickens and the subsequent resistance to ZIKV infection.

While the differential innate immune response of the two groups compared in this study may be impacted by the differences in the magnitude of the ZIKV input, the innate response of the neonatal chick immune system is underdeveloped compared with 4- to 6-week-old chickens [97] and is probably the greatest factor causing the lack of interferon and IL1β gene activation in 1-day-old chicks following ZIKV exposure. The single upregulated gene observed in 1-day-old chickens was OASL. OASL is induced by multiple viral infections in chicken [98], and studies of WNV infection with recombinant chicken OASL indicate flavivirus-targeted host-restriction activity [99,100]. Possibly, the young chicks cleared the virus through interferon type 2 or 3 responses since OASL can be induced by these mechanisms. These possibilities will require further exploration.

In the present study, the presence of infectious or replicating virus was not assessed in the blood or tissue, so it is possible that the detected ZIKA RNA represents input virus. However, we observed an increase in the maximum viral RNA copy number in the spleen and crop between days 2 and 3 post-inoculation (Figure 4), and no viral RNA was detected in birds that were administered an equivalent number of UV-inactivated ZIKV particles. Another limitation of the present study was the lack of perfusion to remove blood before tissue collection. Therefore, the ZIKV RNA detected in the tissues may represent virus present in the blood. At later timepoints, this scenario is unlikely because ZIKV RNA was not observed in the blood plasma beyond 5 dpi. In Experiment 6, no viral RNA was detected in the heart tissue despite observing viral RNA in the plasma at the same timepoint (Figure 2F and Table 1). Viral RNA in the heart tissue was not assessed in the subsequent experiments 7 and 8. Additionally, several factors that are known to influence ZIKV tropism, pathogenicity, and severity were not tested in this study. These include inoculation using vector-generated virus (i.e., mosquito bite or virus propagated in mosquito cells) and injection at other body sites [76,101,102,103].

In summary, our results strongly suggest that chickens do not have the potential to play a role in the ZIKV transmission cycle, as either a reservoir or an accidental host. Additionally, ZIKV does not appear to present a health threat to chickens or individuals closely interacting with poultry. Chickens would likely be poor sentinels for ZIKV since ZIKV-specific antibody levels were only found in young animals who received high amounts of input virus, and even these were quite low, which is consistent with previous reports [65,66].

## Figures and Tables

**Figure 1 viruses-16-00569-f001:**
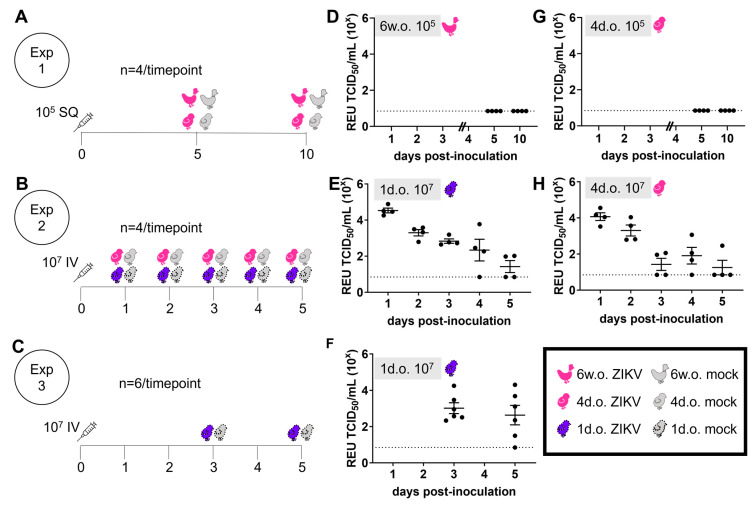
Plasma viremia of chickens inoculated with Zika virus. Chickens of varying ages (6 weeks old (w.o.), 4 days old (d.o.) and 1 (d.o.) were experimentally inoculated subcutaneously (SQ) or intravenously (IV) with varying TCID_50_ doses of Zika virus (ZIKV) or vehicle (mock). (**A**–**C**) Experimental design of three separate infection experiments. (**D**–**H**) ZIKV viral RNA in plasma from animals euthanized at selected timepoints was quantified by real-time RT-PCR. Relative equivalent units (REUs) from a TCID_50_ standard are shown for individual animals from Experiment (Exp) 1 (**D**,**G**), Exp2 (**E**,**H**), and Exp3 (**F**). Dashed lines indicate the limit of detection, horizontal lines represent mean, and vertical error bars indicate standard error of the mean.

**Figure 2 viruses-16-00569-f002:**
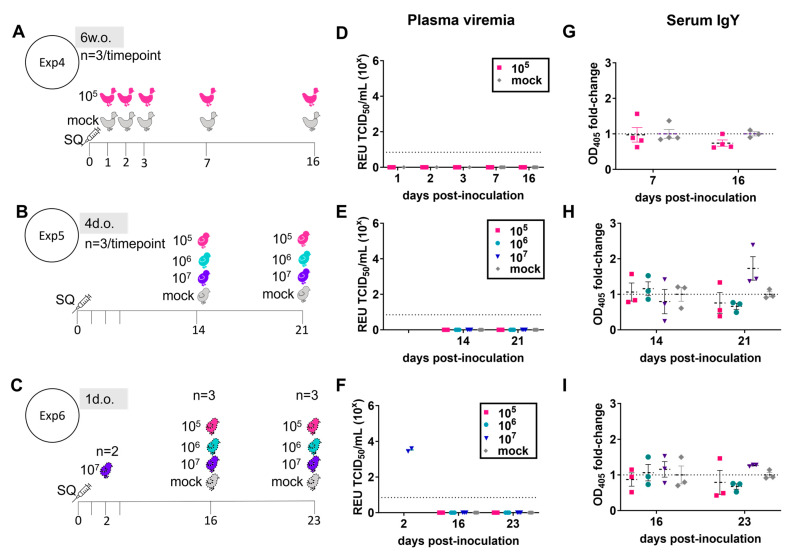
Viremia and antibody response in chickens following inoculation with Zika virus. Chickens aged 6 weeks (6 w.o.), 4 days (4 d.o.), or 1 day (1 d.o.) were inoculated subcutaneously with vehicle (mock) or varying TCID_50_ doses of Zika virus (ZIKV) and euthanized at selected timepoints. (**A**–**C**) Experimental design of three separate infection experiments. (**D**–**F**) ZIKV viral RNA in plasma, quantified by real-time PCR, are shown for individuals in Experiment (Exp) 4 (**D**), Exp5 (**E**), and Exp6 (**F**) as relative equivalent units (REUs) against a TCID_50_ standard. Dashed lines indicate the limit of detection. (**G**–**I**) ZIKV-specific IgY antibody levels quantified as fold-change in optical densities at 405 nm (OD_405_) over mean mock-infected values (dashed line) for Exp4 (**G**), Exp5 (**H**), and Exp6 (**I**) are shown for individuals. Horizontal lines represent mean, and vertical whiskers indicate standard error of the mean.

**Figure 3 viruses-16-00569-f003:**
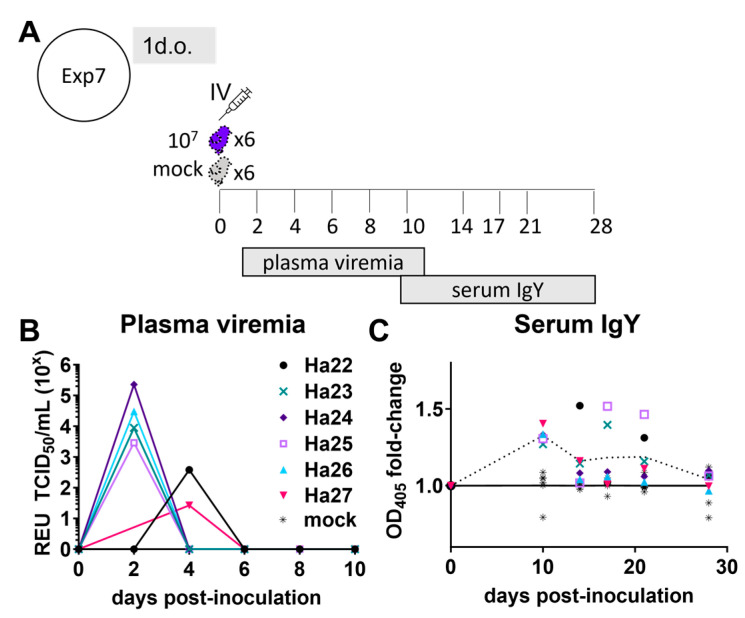
Longitudinal viremia and antibody response in 1-day-old (1 d.o.) chickens inoculated with Zika virus. Chickens were inoculated with 10^7^ TCID_50_ units of Zika virus (ZIKV), and blood was collected for quantification of plasma viremia (**B**) and/or serum anti-ZIKV IgY antibodies at specified timepoints. (**A**) the experimental design of this experiment (Exp7). (**B**) ZIKV viral RNA in plasma quantified by real-time PCR relative equivalent units (REU) against a TCID_50_ standard are shown for ZIKV-inoculated animals Ha22-27 individually and mock-infected animals as mean. (**C**) ZIKV-specific IgY antibody levels quantified as fold-change in optical densities at 405 nm (OD405) over mean mock-infected values (dashed line). Fold-change values are shown for all individuals in ZIKV- and mock-inoculated groups. Dashed line represents mean fold-change value of ZIKV-inoculated individuals.

**Figure 4 viruses-16-00569-f004:**
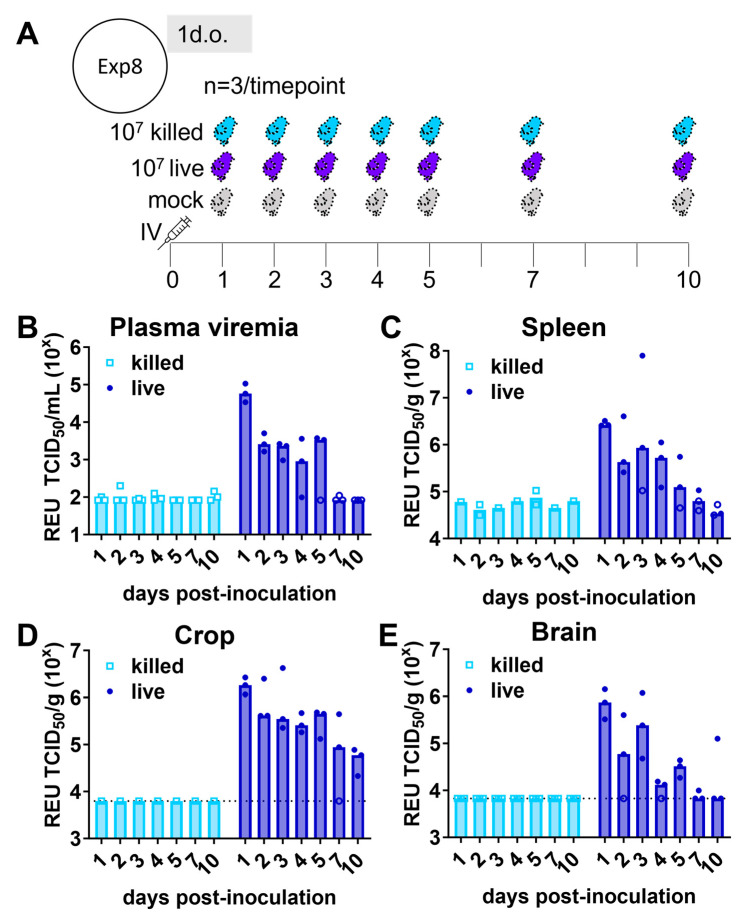
Viral RNA detection in tissues of 1-day-old (1 d.o.) chickens inoculated with live or killed Zika virus. Chickens were inoculated with 10^7^ TCID_50_ units of infectious Zika virus (ZIKV; live) or UV-inactivated ZIKV (killed) or vehicle (mock), and blood and tissues were collected following euthanasia at selected timepoints. (**A**) The experimental design of this experiment (Exp8). (**B**–**D**) ZIKV viral RNA quantified by real-time PCR relative equivalent units (REUs) against a TCID_50_ standard are shown. The median of animals per timepoint is shown by height of bar graph. Symbols represent measurements from individual animals in plasma (**B**), spleen (**C**), crop (**D**), and brain (**E**). Open symbols represent individuals with undetectable viral RNA, at or below the limit of detection (LOD); the position of the symbol indicates the LOD. In B and C, LOD was not uniform across all specimens due to volume or mass variability. In D and E, LOD was equivalent across all specimens and is shown as a dashed line. All specimens tested from the killed inoculum group had undetectable viral RNA.

**Figure 5 viruses-16-00569-f005:**
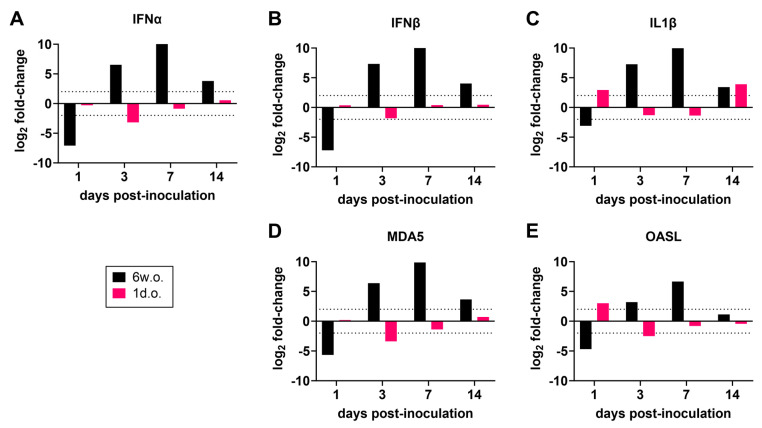
Gene expression of chickens inoculated with Zika virus. RNA of selected genes was measured against RNA from 18S by real-time PCR using tissue homogenates of 6-week-old (6 w.o.) and 1-day-old (1 d.o.) chickens inoculated with Zika virus (ZIKV) or mock-inoculated. Genes evaluated were (**A**) interferon alpha (IFNα), (**B**) interferon beta (IFNβ), (**C**) interleukin 1 beta (IL1β), (**D**) melanoma differentiation-associated protein 5 (MDA5), and (**E**) 2′-5′-oligoadenylate synthetase-like protein (OASL). Relative expressions of ZIKV-inoculated compared to mock-inoculated chickens are shown, with height of bar indicating mean of 2 to 4 animals per timepoint. The 6 w.o. specimens were from Exp4 (Figure 2); 1 d.o. day 16 post-inoculation specimens were from Exp6 (Figure 2), and day 1–7 specimens were from Exp8 (Figure 4).

**Table 1 viruses-16-00569-t001:** Viral RNA detection in tissues at 2 days post-inoculation of 1-day-old chicks inoculated with 10^7^ TCID_50_ units of Zika virus.

**Tissue**	**Chick A**	**Chick B**
crop ^a^	7.38 × 10^5^	1.72 × 10^5^
liver	5.20 × 10^4^	3.16 × 10^4^
brain	1.45 × 10^4^	-
kidney	8.42 × 10^3^	-
spleen	4.71 × 10^3^	-
lung	- ^b^	-
heart	-	-
duodenum	-	-
eye	-	-

^a^ Limit of detection in relative equivalent TCID_50_ units/gram: brain, crop, eye, and duodenum: 1.53 × 10^3^; liver, lung, and heart: 3.07 × 10^3^; kidney: 1.39 × 10^3^; spleen: 2.55 × 10^3^. ^b^ indicates no viral RNA detected.

## Data Availability

Dataset available on request from the authors.

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
