# Peer review of "The Susceptibility of Chickens to Zika Virus: A Comprehensive Study on Age-Dependent Infection Dynamics and Host Responses"

_viruses, 2024, doi:10.3390/v16040569_

Round 1

Reviewer 1 Report

Comments and Suggestions for Authors

General Comments

The main issue with this study is that inoculation was IV or SQ with a virus that was propagated in mammalian cells. There has been a great deal of research that shows that inoculation route, the presence of vector saliva and the cell line (vector vs host) used for virus propagation makes a difference in infection and disease phenotype. Yet, this research is not acknowledged in this study.

Please check that all abbreviations are defined at first use and are consistent throughout the manuscript.

Specific Comments

Introduction

Lines 63-71 - The argument that the presence of chickens and the mosquitoes that feed on them may increase the possibility of Zika virus transmission to humans needs to be more fully developed. What is the competence of ornithophilic mosquitoes for Zika virus? How often do Aedes aegypti and Ae. albopictus feed on birds when mammals/humans are present?

Materials and Methods

Line 127 - 32000 x g?

Host Gene PCR - isn't this qPCR rather than PCR?

Viral PCR - shouldn't this be qRT-PCR?

Results

The text does not match Figure 1.

Line 224 - dpi not defined at first use.

Line 225 - do not defined at first use.

Since tissues were evaluated by qRT-PCR, isn't it possible that the results are from blood in the tissues rather than infection of the tissues themselves?

Line 284 - Was this sample tested again? Was the extraction equal to the other samples?

Line 285 - What was the level needed to achieve seroconversion? Most appear to have little difference from mock infected. For the chicks with the highest fold-changes, was a neutralization assay done to see if the antibodies had any effect?

Line 299 - This is such a small sample size that saying there is a productive infection is tenuous at best. Follow up with another technique such as IHC would bolster the interpretation.

Figure 4 - Limit of detection needed in panels B and C.

Author Response

Thank you for your valuable feedback. We have revised the manuscript to reflect our research study better and address the comments provided. We appreciate your input and hope you find the revisions satisfactory.

General Comments

The main issue with this study is that inoculation was IV or SQ with a virus that was propagated in mammalian cells. There has been a great deal of research that shows that inoculation route, the presence of vector saliva and the cell line (vector vs host) used for virus propagation makes a difference in infection and disease phenotype. Yet, this research is not acknowledged in this study.

Thank you for this useful feedback. In the revised manuscript we include this limitation of the study in the Discussion, which reads “Additionally, several factors that are known to influence ZIKV tropism, pathogenicity, and severity were not tested in this study. These include inoculation using vector-generated virus (i.e. mosquito bite or virus propagated in mosquito cells) and injection at other body sites.” 

Please check that all abbreviations are defined at first use and are consistent throughout the manuscript.

Thank you for your comment. We have thoroughly reviewed the manuscript to ensure that all abbreviations are defined upon first use and are consistently used throughout the document.

Specific Comments

Introduction

Lines 63-71 - The argument that the presence of chickens and the mosquitoes that feed on them may increase the possibility of Zika virus transmission to humans needs to be more fully developed. What is the competence of ornithophilic mosquitoes for Zika virus? How often do Aedes aegypti and Ae. albopictus feed on birds when mammals/humans are present?

Our intent here primarily was to describe a possible mode for reverse zoonotic transmission of Zika virus to chickens (not from chickens to humans), although we recognize the reviewer’s interest in the zoonotic transmission potential also – if chickens were in fact susceptible. We have edited this paragraph for clarification and now include examples of studies in which Aedes aegypti or Aedes albopictus obtain bloodmeals from chickens while also having access to mammals.

Materials and Methods

Line 127 - 32000 x g?

This has been corrected to “3,200 x g.”

Host Gene PCR - isn't this qPCR rather than PCR?

This is now changed from “PCR” to “qPCR.”

Viral PCR - shouldn't this be qRT-PCR?

This is now changed from “PCR” to “qRT-PCR.”

Results

The text does not match Figure 1.

The updated manuscript now contains accurate references to the Figure 1 panels in the text. Thank you for noting this.

Line 224 - dpi not defined at first use.

This been corrected so that the first use of “dpi” defines it as “days post-inoculation” in line 226.

Line 225 - do not defined at first use.

This line was corrected to replace “do” with “days old.” “Days old” is used throughout the remainder of the manuscript.

Since tissues were evaluated by qRT-PCR, isn't it possible that the results are from blood in the tissues rather than infection of the tissues themselves?

We agree that this is a possibility. An assessment is included in the revised manuscript in Lines 913-919, which reads:

Another limitation of the present study was the lack of perfusion to remove blood before tissue collection. Therefore, viral RNA detected in tissues may represent virus present in the blood. At later timepoints, this scenario is unlikely because did not observe viral RNA in blood plasma beyond 5 dpi. In experiment 6, no viral RNA was detected in heart tissue despite observing viral RNA in plasma at the same timepoint. Heart tissue was not assessed subsequent experiments 7 and 8.”

Line 284 - Was this sample tested again? Was the extraction equal to the other samples?

Thank you for raising this question. We hope the reviewer can understand how difficult it can be to obtain blood products from young chicks, particularly when the serial collection places further ethical restrictions on the volume collected. Therefore, we did not have excess plasma to re-extract and re-test. However, when we reviewed the raw data, we verified your suspicion that this sample did have a lower volume than the others. We have added information about this in lines 653-656. The section now reads:

Several chicks had peak viremia at 2 days post-inoculation (Figure 3B). The volume of plasma obtained and tested from chick Ha-22 on 2 dpi was 67-88% less than from other chicks, which may explain why this chick had undetectable virus at 2 dpi but detectable virus at 4 dpi (Figure 3B).”

Line 285 - What was the level needed to achieve seroconversion? Most appear to have little difference from mock infected. For the chicks with the highest fold-changes, was a neutralization assay done to see if the antibodies had any effect?

We re-worded this statement for more accuracy and removed the term “seroconverted.” Now it says: “On one or more days, serum ZIKV-specific IgY from all ZIKV-inoculated chicks in this tracked cohort was at least 20% higher than that in all mock-inoculated chicks at the same timepoint. Average fold-change of anti-ZIKVirus serum IgY from ZIKV-infected chicks was significantly higher than from mock-infected controls through 21 dpi (Figure 3).”

In the course of the ELISA assay development, we validated our assay with that from another lab using a subset of the serum samples. The other lab also assayed the samples with their PRNT assay, but no neutralizing antibodies were identified. We do not believe that the weak antibody levels found by ELISA contain any neutralizing ability. The inter-lab comparison is now included in the Methods (lines 541-544) to clarify this.

Line 299 - This is such a small sample size that saying there is a productive infection is tenuous at best. Follow up with another technique such as IHC would bolster the interpretation.

We agree that each experiment used a small number of animals at each timepoint and that confirmation using a secondary technique would be beneficial. Unfortunately, we were not able to obtain tissue for those techniques. We acknowledge this weakness of the study in the revised Discussion (lines 897-910). Additionally, we changed the word “evidence” to “indication” in line (672) to indicate the reduced strength of the finding described in that paragraph.

Figure 4 - Limit of detection needed in panels B and C.

Information about the limit of detection (LOD) in panels B and C of Figure 4 is included in the Figure Legend. Since the volume or mass of the specimens tested in these panels varied because the specimens are very small, the limit of detection was unique to each sample. For this reason, samples with undetectable ZIKV RNA in the qRT-PCR assay are represented by open symbols to indicate that this is the LOD for that specimen. Measurements above the LOD are represented with closed symbols. Figure 4 is modified in the revised manuscript to include color, which we hope will increase visibility of the closed and open symbols. The Legend text that describes the LOD reads: “Open symbols represent individuals with undetectable viral RNA, at or below the limit of detection (LOD); the position of the symbol indicates the LOD. In B and C, LOD was not uniform across all specimens due to volume or mass variability. In D and E, LOD is equivalent across all specimens and is shown as a dashed line.”

Reviewer 2 Report

Comments and Suggestions for Authors

An interesting paper, even if somewhat negative in terms of Zika infecting chickens and repeating, in part, previous work (refs 45, 46). 

Use of colour in the b/w figs would really help illustrate the findings better.

L37 Unclear what is being alluded to here regarding chicken STAT2 – is this like mouse STAT2 and not destroyed by Zika NS5?  Or is nothing known and this is speculation? The paper does not provide any information regarding chicken STAT2 – so not clear why this is in the Abstract.  Are there structure function similarities that point to this contention? There is also the issue of STAT1, which could equally be involved here (Sci Adv. 2022 Nov; 8(48): eadd8095).  Some actual STAT2 data (homology?) would greatly add to this paper, but at this stage is surely only a discussion point.

L49 “gained prominence” is an odd phrase here when talking about damaged neonates. Also should talk about congenital Zika syndrome (CZS) – not just microcephaly.  CZS is a much broader problem with long term health and development implications.

L60 Actually not sure that this statement is correct. African viruses are more neurovirulent and may thus induce miscarriage and/or fetal demise, thereby reducing the likelihood of births of viable infants with CZS (PLoS Pathog 17(7): e1009788)

L62 Its mosquitoes

L65 for not to.

L72 Evidence is a noun.

L89 to 94 has no refs?

L79 Not just  microcephaly but CZS.

L84 although very rare direct

L89 Neonatal mice are not really immunocompetent

L 330 Should be made clear that this may simply be movement of immune cells out of the spleen rather than down regulation of gene expression by cells in the spleen.  Unless spleen populations are monitored one cannot conclude down regulation of gene expression by cells in the spleen.  This is the time leukocyte mobilisation might be expected.

L351 “prevalent”? This should be 'present' surely – surely not prevalent?

The Discussion needs some work and could be much shorter; the first paragraph simply repeats the Introduction, the second Results, the third l370-380 is also introduction, and l390-399 is again Results repeated.

Comments on the Quality of English Language

Mostly fine

Author Response

Thank you for your valuable feedback. We have revised the manuscript to reflect our research study better and address the comments provided. We appreciate your input and hope you find the revisions satisfactory.

An interesting paper, even if somewhat negative in terms of Zika infecting chickens and repeating, in part, previous work (refs 45, 46).

Use of colour in the b/w figs would really help illustrate the findings better.

The Figures 1-4 and Supplementary Figure S1 have been updated to include color. Now no figures are solely black and white.

L37 Unclear what is being alluded to here regarding chicken STAT2 – is this like mouse STAT2 and not destroyed by Zika NS5?  Or is nothing known and this is speculation? The paper does not provide any information regarding chicken STAT2 – so not clear why this is in the Abstract.  Are there structure function similarities that point to this contention? There is also the issue of STAT1, which could equally be involved here (Sci Adv. 2022 Nov; 8(48): eadd8095).  Some actual STAT2 data (homology?) would greatly add to this paper, but at this stage is surely only a discussion point.

Thank you for your valuable feedback and highlighting the need for further clarification regarding chicken STAT2 in our manuscript. We appreciate your suggestion to include additional data and insights to address this concern. In response, we have provided a supplementary figure (Supplementary Figure S2) that offers a comprehensive comparison of STAT2 proteins between chicken, human, and mouse. This figure includes maximum likelihood phylogeny analysis, multiple sequence alignment of amino acids, and structural alignment of human, mouse, and chicken STAT2. We believe this additional information will clarify the differences and potential implications of chicken STAT2 in relation to Zika NS5 interaction.

L49 “gained prominence” is an odd phrase here when talking about damaged neonates. Also should talk about congenital Zika syndrome (CZS) – not just microcephaly.  CZS is a much broader problem with long term health and development implications.

The sentence has been edited and now reads “Initially identified in 1947, intensified scrutiny of the virus followed a surge in microcephaly cases associated with a 2015 ZIKV outbreak in Brazil.” Specific mention of CZS is now included in the next sentence (line 219) and is further discussed in the Introduction (lines 251-256).

L60 Actually not sure that this statement is correct. African viruses are more neurovirulent and may thus induce miscarriage and/or fetal demise, thereby reducing the likelihood of births of viable infants with CZS (PLoS Pathog 17(7): e1009788)

We have removed this statement from the manuscript, and we agree that the original report about African and Asian lineages in our original manuscript was not the whole story about those two lineages.

L62 Its mosquitoes

The term “an arthropod vector” has been replaced with “a mosquito” in this sentence.

L65 for not to.

The word “to” has been replaced with “for” in this sentence.

L72 Evidence is a noun.

The word “evidence” has both noun and verb forms in Oxford and Merrian-Webster dictionaries. For clarification. the word “evidenced” has been replaced with “suggested” in this sentence.

L89 to 94 has no refs?

References have been added.

L79 Not just  microcephaly but CZS.

CZS is now included here (revised manuscript Lines 251-256), with a description of the syndrome and its consequences.

L84 although very rare direct

The phrase “very rare” has been added to this sentence.

L89 Neonatal mice are not really immunocompetent

Thank you for highlighting this detail. The sentences and references have been updated to clarify that it is the pups born from immunocompetent mice that are susceptible. This section now reads:

“Other immunocompetent animals shown to support ZIKV infection include some guinea pigs and Syrian golden hamsters. Fetuses and newborn pups of immunocompetent mice are also susceptible.”

L 330 Should be made clear that this may simply be movement of immune cells out of the spleen rather than down regulation of gene expression by cells in the spleen.  Unless spleen populations are monitored one cannot conclude down regulation of gene expression by cells in the spleen.  This is the time leukocyte mobilisation might be expected.

We appreciate this insight. The sentence has been modified to reflect this possibility and now reads: “In the spleen of chickens infected at 6-weeks of age, the initial response at 1 dpi was characterized by reduced expression of antiviral genes OAS-L and MDA5 and type-1 interferons IFN-β and IFN-α (Figure 5), which could represent migration of immune cells from the spleen to peripheral site.”

L351 “prevalent”? This should be 'present' surely – surely not prevalent?

The world “prevalent” has been replaced with “present” in this sentence.

The Discussion needs some work and could be much shorter; the first paragraph simply repeats the Introduction, the second Results, the third l370-380 is also introduction, and l390-399 is again Results repeated.

Thank you for this suggestion for improving our manuscript. The Discussion section has been revised to remove repetitive material as noted by the Reviewer and to include limitations of the study.

Reviewer 3 Report

Comments and Suggestions for Authors

The terminology used in some of the paragraphs can be improved.

Overall this is a well-written paper that describes some important work and builds on previous observations that chickens probably play an insignificant role in the epidemiology of this disease.

Comments on the Quality of English Language

Line 26

The word “virus” is a noun so the appropriate term is viral epidemiology not virus epidemiology

Line 29

Symptoms are described by a human patient. Surely these chickens are not able to carry on a conversation. The authors probably meant clinical signs.

Line 30

The authors indicate that for day old chicks exhibited viraemia. While a viraemia was observed the chicks do not actively participate in exhibiting the viraemia. Samples are collected and tested subsequently it is deduced that a viraemia has occurred.

Line 34

The chickens do not show a type I interferon response. The authors could say that a response was observed.

Line 35

The chicks do not display a limited innate immune activation. This needs to be rewritten to indicate that this was measured.

Line 48

The two references are reviews of Zika virus infection. The authors could also have used (Hasan et al., 2019). However, none of these references are definitive taxonomic papers. They could probably have best been used at the end of the paragraph. If the taxonomy of the organism is the major focus then it is important to use a taxonomy reference. As they have been some recent changes to the approach of the taxonomy of the family Flaviviridae the reference (Postler et al., 2023) should also be included.

Paragraph 2.7 Viral PCR

Was the kit that was used SuperScript™ III Platinum™ SYBR™ Green One-Step qRT-PCR Kit?. As the cycling parameters were described in paragraph 2.6 it is a little strange that the cycling parameters for this assay were not described. If SYBR green was used did the authors use a high-resolution melt curve to confirm the specificity of the reaction?

Line 203

The chickens do not actively participate in displaying a viraemia. Rewrite the sentence.

Line 225

The chickens do not actively participate in demonstrating Zika virus in their plasma. Plasma samples were collected and tested and subsequently 50% of the samples reacted in the RT-PCR.

Line 230

The chickens do not describe the clinical symptoms. Clinical signs may have been observed. Similar comments apply to lines 236 and 240.

Line 247

Tissue samples do not demonstrate the presence of virus. An assay was carried out and a result was generated.

Line 279

The authors indicate that virus levels detected in plasma demonstrated a peak. Clearly these are viral titres and they do not demonstrate a peak. The peak was observed.

Line 360

These are clinical signs not symptoms.

Line 362

The virus was undetectable in tissue samples or in the samples collected from the birds.

HASAN, S., SAEED, S., PANIGRAHI, R. & CHOUDHARY, P. 2019. Zika Virus: A Global Public Health Menace: A Comprehensive Update. J Int Soc Prev Community Dent, 9, 316-327.

POSTLER, T. S., BEER, M., BLITVICH, B. J., BUKH, J., DE LAMBALLERIE, X., DREXLER, J. F., IMRIE, A., KAPOOR, A., KARGANOVA, G. G., LEMEY, P., LOHMANN, V., SIMMONDS, P., SMITH, D. B., STAPLETON, J. T. & KUHN, J. H. 2023. Renaming of the genus Flavivirus to Orthoflavivirus and extension of binomial species names within the family Flaviviridae. Arch Virol, 168, 224.

Author Response

Thank you for your valuable feedback. We have revised the manuscript to reflect our research study better and address the comments provided. We appreciate your input and hope you find the revisions satisfactory.

The terminology used in some of the paragraphs can be improved.

Thank you for this suggestion. Based on these and other specific suggestions from reviewers and our own re-reading of the manuscript, updates to the terminology are included in the revised manuscript. These include replacing “clinical symptoms” (and other versions of the word “symptom”) with “clinical signs”, including Congenital Zika Syndrome as a crucial result of Zika virus infection, ensuring that results are described without using chickens as the grammar-based subject, and using specific assay descriptions for PCR-based methods headings.

Overall this is a well-written paper that describes some important work and builds on previous observations that chickens probably play an insignificant role in the epidemiology of this disease.

Comments on the Quality of English Language

Line 26. The word “virus” is a noun so the appropriate term is viral epidemiology not virus epidemiology

Thank you to this reviewer for the great suggestions for improving the writing quality. In this line, the word “virus” has been replaced with “viral”.

Line 29. Symptoms are described by a human patient. Surely these chickens are not able to carry on a conversation. The authors probably meant clinical signs.

We agree. Thank you for catching this error. We haven’t figured out how to communicate with the chickens yet! The word “symptoms” has been changed to “signs” here, and a similar change was made in Line 91 which originally described experimental animal infections as “asymptomatic.”

Line 30. The authors indicate that for day old chicks exhibited viraemia. While a viraemia was observed the chicks do not actively participate in exhibiting the viraemia. Samples are collected and tested subsequently it is deduced that a viraemia has occurred.

This sentence has been edited and now reads: “Although virus was detectable in blood and tissues during the first five days post-inoculation in 1-day and 4-day-old chicks inoculated with a high viral dose, ZIKV was undetectable in 6-week-old birds at all timepoints.”

Line 34. The chickens do not show a type I interferon response. The authors could say that a response was observed.

 This line has been edited to address this problem and now reads: “Innate immune responses varied significantly between age groups. Robust type I interferon and inflammasome responses were measured in older chickens, while limited innate immune activation was observed in younger chicks.”

Line 35. The chicks do not display a limited innate immune activation. This needs to be rewritten to indicate that this was measured.

This line has been edited to address this problem and now reads: “Innate immune responses varied significantly between age groups. Robust type I interferon and inflammasome responses were measured in older chickens, while limited innate immune activation was observed in younger chicks.”

Line 48. The two references are reviews of Zika virus infection. The authors could also have used (Hasan et al., 2019). However, none of these references are definitive taxonomic papers. They could probably have best been used at the end of the paragraph. If the taxonomy of the organism is the major focus then it is important to use a taxonomy reference. As they have been some recent changes to the approach of the taxonomy of the family Flaviviridae the reference (Postler et al., 2023) should also be included.

 Thank you for noting this error. The revised manuscript now includes the latest ICTV nomenclature with the appropriate reference in lines (44-45). As suggested, references relating to the overall Zika virus history have been moved to the end of the paragraph and include Hasan et al., 2019 (lines 226).

Paragraph 2.7 Viral PCR. Was the kit that was used SuperScript™ III Platinum™ SYBR™ Green One-Step qRT-PCR Kit?. As the cycling parameters were described in paragraph 2.6 it is a little strange that the cycling parameters for this assay were not described. If SYBR green was used did the authors use a high-resolution melt curve to confirm the specificity of the reaction?

Thank you for noting that some information is missing with this method. The RT-PCR was a probe-based approach. We now state that there is a probe and include the cycling conditions. This section now includes:

“Detection of ZIKV RNA from extracted viral RNA from plasma, tissue samples, and swabs was performed using SuperScript III Platinum One-Step qRT-PCR Kit on an Applied Biosystems 7500 Fast Real-Time PCR machine on extracted viral RNA from plasma, tissue samples, and swabs using primers and probe listed in Supplementary Table S2 [1]. The RT-PCR reaction was run at 50°C for 30 min, 95°C for 2 min, and 45 cycles of 95°C for 15 s and 55°C for 30 s. . . .”

Line 203. The chickens do not actively participate in displaying a viraemia. Rewrite the sentence.

 This sentence now reads “At 5 and 10 days following inoculation, no plasma viremia was detected in the chickens . . . “

Line 225. The chickens do not actively participate in demonstrating Zika virus in their plasma. Plasma samples were collected and tested and subsequently 50% of the samples reacted in the RT-PCR.

This sentence has been modified to now read: “On each of the five days after inoculation into 1-day old chicks, ZIKV RNA was detectable in the plasma of 50% or more of tested chicks  (Figure 1E).”

Line 230. The chickens do not describe the clinical symptoms. Clinical signs may have been observed. Similar comments apply to lines 236 and 240.

 “Symptoms” has been changed to “signs” in each of these lines.

Line 247. Tissue samples do not demonstrate the presence of virus. An assay was carried out and a result was generated.

 This sentence has been edited and now reads “At 2 dpi, ZIKV RNA was detected in crop, liver, brain, kidney and spleen from 1-day old chicks, but not in duodenum, heart, lung or eye tissue from the same individuals (Table 1).”

Line 279. The authors indicate that virus levels detected in plasma demonstrated a peak. Clearly these are viral titres and they do not demonstrate a peak. The peak was observed.

 The sentence has been modified to begin as follows: “A peak in ZIKV RNA was observed at day 1 after inoculation followed by a subsequent decline, . . .”

Line 360. These are clinical signs not symptoms.

 “Symptoms” changed to “clinical signs.”

Line 362. The virus was undetectable in tissue samples or in the samples collected from the birds.

 Sentence has been modified to read “Virus was undetectable in tissue and blood samples from 6-week old birds at all timepoints (1 to 16 dpi).”

  1. Goebel, S., et al., A sensitive virus yield assay for evaluation of antivirals against Zika Virus. Journal of Virological Methods, 2016. 238: p. 13-20.
